

# Evolutionary analysis of vision genes identifies potential drivers of visual differences between giraffe and okapi

Edson Ishengoma[1,2], Morris Agaba[1] and Douglas R. Cavener[3]

[1] The School of Life Sciences and Bio-Engineering, Nelson Mandela African Institution of Science and Technology, Arusha, Tanzania
[2] Mkwawa University of College of Education, University of Dar-es-Salaam, Iringa, Tanzania
[3] Department of Biology and the Huck Institute of Life Sciences, Pennsylvania State University, University Park, PA, United States

## ABSTRACT

**Background**. The capacity of visually oriented species to perceive and respond to visual signal is integral to their evolutionary success. Giraffes are closely related to okapi, but the two species have broad range of phenotypic differences including their visual capacities. Vision studies rank giraffe's visual acuity higher than all other artiodactyls despite sharing similar vision ecological determinants with many of them. The extent to which the giraffe's unique visual capacity and its difference with okapi is reflected by changes in their vision genes is not understood.

**Methods**. The recent availability of giraffe and okapi genomes provided opportunity to identify giraffe and okapi vision genes. Multiple strategies were employed to identify thirty-six candidate mammalian vision genes in giraffe and okapi genomes. Quantification of selection pressure was performed by a combination of branch-site tests of positive selection and clade models of selection divergence through comparing giraffe and okapi vision genes and orthologous sequences from other mammals.

**Results**. Signatures of selection were identified in key genes that could potentially underlie giraffe and okapi visual adaptations. Importantly, some genes that contribute to optical transparency of the eye and those that are critical in light signaling pathway were found to show signatures of adaptive evolution or selection divergence. Comparison between giraffe and other ruminants identifies significant selection divergence in *CRYAA* and *OPN1LW*. Significant selection divergence was identified in *SAG* while positive selection was detected in *LUM* when okapi is compared with ruminants and other mammals. Sequence analysis of *OPN1LW* showed that at least one of the sites known to affect spectral sensitivity of the red pigment is uniquely divergent between giraffe and other ruminants.

**Discussion**. By taking a systemic approach to gene function in vision, the results provide the first molecular clues associated with giraffe and okapi vision adaptations. At least some of the genes that exhibit signature of selection may reflect adaptive response to differences in giraffe and okapi habitat. We hypothesize that requirement for long distance vision associated with predation and communication with conspecifics likely played an important role in the adaptive pressure on giraffe vision genes.

Corresponding author
Edson Ishengoma,
ishengomae@nm-aist.ac.tz,
edson.ishengoma@muce.ac.tz

## INTRODUCTION

Visual cognition is critical to health, survival and evolutionary success of terrestrial vertebrates. Visual perception in mammals is attributable to various aspects such as visual acuity, contrast sensitivity, motion perception, depth perception and color discrimination (*Osorio & Vorobyev, 2005*; *Kohn, 2007*; *Heesy & Hall, 2010*). These visual properties are inextricably linked to species evolutionary fitness. They contribute to species competitiveness at food acquisition, predator avoidance, suitable mate recognition, intra-specific communication and locating suitable habitat (*Horth, 2007*; *Tisdale & Fernandez-Juricic, 2009*; *Dimitrova & Merilaita, 2010*).

Vision and ecological studies appear to show that considerable differences in visual perception exist between giraffe and other artiodactyls including its close relative, the okapi. Giraffes predominantly rely on visual communication due to their excellent aerial vision (*Young & Isbell, 1991*; *VanderWaal et al., 2013*; *Veilleux & Kirk, 2014*). This is partly mediated by their uniquely long legs and neck in comparison to other ruminants (*Mitchell et al., 2013*). But okapi exhibit poor eyesight and mostly rely on acute sense of smell and hearing to exploit their low light environment (*Lindsey, Green & Bennett, 1999*; *Greive & Iwago, 2003*). Since giraffes inhabit the open Savannah while okapi are restricted to dense Congo forests, differences in vision between giraffe and okapi could be a function of their ecologies. However, the basis of giraffe's uniquely excellent vision even among other artiodactyls with overlapping ecological niches remains enigmatic.

Although gross morphological and cellular aspects of the eye can contribute to vision differences between species (*Pettigrew, 1986*; *Jeon, Strettoi & Masland, 1998*; *Lindsey, Green & Bennett, 1999*; *Sivak, Andison & Pardue, 1999*; *Cepko, 2014*), adaptive variation in vision traits will ultimately be determined by changes at the gene level. Vision in vertebrates is mediated by photoreceptors (rods and cones) which contain visual pigments (opsins) bound to a retinal chromophore (*Wald, 1935*). Five classes of opsins are expressed in vertebrate photoreceptors: a rod class of pigment RH1 (rhodopsin) enables animals to see in dim light, and four classes of cone pigments which enable perception of distinct color wavelengths. The cone class of pigments include short wavelength sensitive type 1 opsin (SWS1/OPN1SW/S) sensitive in the ultraviolet/violet range of the spectrum at 355–440 nm, SWS type 2 opsin (SWS2) sensitive in the blue range from about 410–490 nm, RH2 opsin (close homolog of RH1) sensitive in the green range from about 480–535 nm, long wavelength opsin (LWS/OPN1LW/L) and middle wavelength-sensitive opsin (MWS/OPN1MW/M) sensitive in the red range from about 490–570 nm (*Yokoyama, 2002*; *Bowmaker, 2008*).

The cloning of bovine rhodopsin gene by *Nathans & Hogness (1983)* inspired efforts to understand variations in molecular genetic mechanisms underlying vision in mammals and vertebrates in general. Comparison of different classes of opsin genes among vertebrates reveals gene loss, gene duplication and nucleotide substitutions to play a fundamental role in the evolution of color vision in vertebrates (*Yokoyama & Radlwimmer, 1998*; *Yokoyama, 2002*; *Horth, 2007*; *Jacobs, 2009*). All mammals have apparently lost RH2 cones while SWS2 is retained only in some egg-laying mammals (*Hunt et al., 2009*). The *LWS* gene duplication

and diversification has resulted into primates possessing both *LWS* and *MWS* genes which, respectively, express L-cone pigment maximally sensitive at around 563 nm and M-cone pigment maximally sensitive at around 535 nm (*Bowmaker, 2008*). This provides some primates with trichromatic vision due to presence of three spectrally distinct cone pigments expressed by *SWS1*, *MWS* and *LWS* genes (*Bowmaker, 2008*). But most eutherian mammals remain dichromatic with *SWS1* and either *MWS* or *LWS* genes (*Collin et al., 2009*).

With respect to the functional mechanism of opsins, spectral tuning in vertebrates is mainly determined by particular amino acids in the opsin protein structure. Two decades ago, *Yokoyama & Radlwimmer (1998)* proposed the "five-sites" rule by demonstrating that sequence changes at sites 180, 197, 277, 285 and 308 were very important in determining variation in *LWS* spectral sensitivity among mammals. Such sequence variations in visual pigments also occur naturally within species, resulting in spectrally variant subtypes of cone pigments among populations with normal color vision. For example, normal color human subjects show 4–5 nm variations based on whether they possess a Serine or Alanine at position 180 of *LWS* (*Merbs & Nathans, 1992*; *Kraft, Neitz & Neitz, 1998*). Consequently, there is an interest to determine whether these inter- and intra-species spectral variations in cone pigments confer visual adaptations in species. Several studies on cichlids and a recent work on New World primates suggest that changes in coding sequence of visual pigments may be associated with matching photoreceptor spectral sensitivity to the visual environment of the respective species (*Hofmann et al., 2009*; *Sabbah et al., 2010*; *Matsumoto et al., 2014*).

While the evolution of opsins and other proteins in the visual phototransduction system has been studied extensively (*Larhammar, Nordström & Larsson, 2009*; *Invergo et al., 2013*), little attention has been given to proteins involved in other processes that impact on whole vision process. Before reaching photoreceptors, light must pass through the ocular media, consisting of sclera, cornea, lens and the vitreous, and these serve to modify and focus light toward the retina. The structure, transparency and light adjustment ability of the ocular media depends on specific constituent proteins (*Pierscionek & Augusteyn, 1993*; *Winkler et al., 2015*). For instance, the sclera and the cornea are packed with collagen fibrils and proteoglycans which provide structural integrity of cornea. An example is lumican (*LUM*), a low molecular weight leucine-rich proteoglycan with keratan sulfate side chain first identified in the cornea as a regulator for organizing collagen fibers in the cornea (*Blochberger et al., 1992*; *Meek & Knupp, 2015*), but is also present in the sclera and other tissues (*Ying et al., 1997*). The crucial role of *LUM* in visual functions is further demonstrated by growing evidence implicating *LUM* in various ocular defects such as corneal opacity and high myopia (*Chakravarti et al., 1998*; *Chakravarti et al., 2000*; *Chakravarti et al., 2003*; *Austin et al., 2002*). The eye lens contains high concentrations of proteins known as crystallins which determine lens transparency and refractive power, but α-crystallin (*CRYAA*) is a major type (*Nagaraj et al., 2012*). Mutations in *CRYAA* are associated with development of cataracts in humans (*Litt et al., 1998*; *Richter et al., 2008*). With the increasing number of whole-genome sequences of many vertebrates, we can now study a broad range of genes enabling functionally integrated traits.

For such an evolutionarily important trait as vision, the associated genes will often be subject to purifying selection and therefore are expected to be conserved over evolutionary timescales (*Lamb, 2011*). However, we recently published giraffe genome and detected signatures of adaptation in few of its vision-associated coding genes (*Agaba et al., 2016*). These genes included *Peripherin-2* (*PRPH2*) and *Cytochrome P450 family 27* (*CYP27B1*). The *PRPH2* encodes a protein integral to rods and cones and mutations in the gene cause various forms of retininis pigmentosa, pattern dystrophies and macular degenerations (*Keen & Inglehearn, 1996*). The *CYP27B1* codes for an enzyme that hydroxylate Vitamin D and modulate normal calcium and phosphorus homeostasis required for proper development and maintenance of bones. Recently, additional *CYP27B1* functions in relation to vision have been proposed which include participating in pathways that counteract inflammation, angiogenesis, oxidative stress and fibrosis. These may in turn confer protection for various retinopathies such as age-related macular degenerations in mice and humans (*Parekh et al., 2007*; *Morrison et al., 2011*).

In order to elucidate on the evolutionary processes underlying disparity in giraffe and okapi vision, we take advantage of the availability of giraffe and okapi genomes to analyze thirty-six (36) candidate 'visual' genes through comparison with those of closely related species. The objectives are first to identify genes exhibiting signatures of adaptive evolution and/or divergent selection and secondly to relate sequence changes in giraffe and okapi vision proteins to possible changes in visual functions.

## MATERIALS AND METHODS

### Identification of candidate genes

To obtain candidate vision genes, multiple strategies were utilized to identify proteins with linked roles in vision. The initial step involved downloading all cattle protein sequences from ENSEMBL (*Flicek et al., 2012*) and screening for proteins annotated with gene ontology (GO) terms "phototransduction" (GO: 0007601), and "visual perception" (GO: 0007602). We used PANTHER (*Mi, Muruganujan & Thomas, 2013*) to first functionally annotate cattle proteins and then screen for those annotated with GO vision terms. The corresponding cattle nucleotide sequences for cattle vision protein were also obtained from ENSEMBL. Since some GO assignments to proteins are not necessarily based on direct experimental evidence but also rely on other evidences such as sequence similarity, we intended to obtain only those genes with proved functions in vision. To achieve this, searches of the literature for proof of gene involvement in vision was performed based upon at least one of the following criteria: (i) the presence of vision disease-associated mutations in human orthologue; (ii) expression in the eye since genes expressed in a given organ at high levels are likely vital in the development and function of that organ and, (iii) interaction with known visual genes and loss of vision in knockout or sporadic mutant mice. Only genes with at least two references linking to a role in vision based on the above criteria were selected. Orthologous mapping of cattle vision proteins to giraffe and okapi genomes returned sequences corresponding to 36 genes. These were the genes that were ultimately used in the analysis (File S1).

## The lineages, gene sequence alignments and gene trees

Other mammalian taxa were selected on the basis of availability of sequences for the 36 candidate vision genes in the RefSeq dataset of GENBANK (*Benson et al., 2013*) or ENSEMBL. Sequences with questionable protein coding quality status based upon having incomplete coding sequence or presence of internal stop codons were removed. The sequences for giraffe and okapi candidate vision genes were obtained by performing TBLASTN search using cattle proteins against giraffe and okapi genome sequences that were generated as part the giraffe genome project (File S2). Also through TBLASTN searches with cattle vision proteins queries, orthologous nucleotide sequences for all 36 vision genes for the target species were downloaded from NCBI RefSeq mRNA or non-redundant nucleotide database. In case of existence of multiple isoforms for a single gene, the isoform with length similar or closest to giraffe and okapi sequences was selected. This is in recognition of the fact that isoforms with similar or equivalent length are likely evolutionarily conserved with similar function among species (*Villanueva-Canas, Laurie & Alba, 2013*). The final list of species, ENSEMBL identity for cattle sequences, RefSeq accession numbers for sequences/isoforms obtained from NCBI and corresponding length for each coding sequence are provided in File S3.

The coding DNA sequences for each gene were translated to the corresponding protein sequence and sequences with internal termination codons were discarded. The protein sequences were then aligned using MUSCLE release 3.8 (*Edgar, 2004*). Subsequently, the protein sequence alignments were used as guides for the generation of coding sequence alignments for each gene. This procedure was implemented using RevTrans (*Wernersson & Pedersen, 2003*). Phylogenetic trees for each gene were constructed using the HKY85 substitution model of nucleotide evolution and maximum likelihood framework implemented in PhyML Version 3.0 (*Guindon & Gascuel, 2003*). Bootstrapping with 100 replicates was performed to be certain of the robustness of the resulting phylogenies.

## Estimation of the average rates of non-synonymous and synonymous substitutions

In order to examine if overall rates of evolution in vision genes contributed to divergence in vision capabilities between giraffe and okapi, the rates of non-synonymous substitutions per non-synonymous sites (dN) and synonymous substitutions per synonymous sites (dS) were estimated for each branch of the tree using the free ratio model of the codeml program in the PAML package (*Yang, 2007*). The free-ratio model independently estimates dN, dS and dN/dS for each branch by assuming that every branch in a tree has a different evolutionary parameter. This is not a robust statistical test for positive selection but the key parameters obtained may provide important information on the relative strengths of selection among species.

## Identification of genes and amino acid residues under positive selection

To determine adaptive evolution on giraffe and okapi vision genes, signatures of positive selection acting across giraffe and okapi lineages against the background of broad range of mammals was independently assessed for each vision gene. The branch-site test for
positive selection was used to identify genes showing signatures of adaptive evolution. The test applies codon models of evolution using normalized nonsynonymous to synonymous substitution rate ratio ($\omega$ or dN/dS). By assuming that adaptive evolution is episodic, where few species in a phylogeny and few sites in a protein are affected by selection, it is required to hypothesize *a priori* a "foreground" branch expected to have evolved under positive selection (*Zhang, Nielsen & Yang, 2005*). The likelihood scores of branch-site alternative and null selection models based on dN/dS as implemented in CODEML of the PAML package were compared using the likelihood ratio test (LRT). Significant case of positive selection was only assumed if LRT yielded $p < 0.05$ using the chi-squared distribution at one degree of freedom. For genes that were identified to be under significant positive selection, amino acid residues in the protein sequences that were predicted by Bayes empirical Bayes (BEB) approach (*Yang, Wong & Nielsen, 2005*) to belong to the codon class of positive selection on the foreground lineages were identified.

## Clade models analyses of selection divergence

It has recently been observed that visual adaptations can also be contributed by divergent selective pressure on homologous visual genes of ecologically divergent species (*Weadick & Chang, 2012*; *Schott et al., 2014*). To explore whether giraffe and okapi differences in vision could partially be explained by divergent selection on their vision proteins, the two species were independently compared with other ruminants by applying PAML's Clade Model C (CmC) (*Bielawski & Yang, 2004*). The CmC partitions different branches within the phylogeny as "background" and "foreground" as well as existence of three site categories, two of which experience uniform selection across the entire phylogeny (either purifying selection ($0 < \omega_0 < 1$) or neutral evolution ($\omega_1 = 1$)) while the third is allowed to vary between background ($\omega_2 > 0$) and foreground ($\omega_3 > 0$) branches. The recently developed M2a_rel (*Weadick & Chang, 2012*) serves as a useful null model for the CmC. In this analysis, category of genes that contribute to the structural properties of cornea and lens and those that are known to play a direct role in the light signaling function were investigated. Of 36 genes, 20 proteins were identified to belong in that category (File S4). In the genes which showed significant selection divergence, potential significance of selection divergence was assessed by examining sites which had significant Bayes posterior probability ($>0.75$) in the divergent site class between target species (giraffe or okapi) and other ruminants. We assessed these sites for possible functional consequences based on literature review of functional studies.

## RESULTS

### Positive selection pressure within the visual genes of giraffe and okapi

Vision genes, as prescribed by gene ontology (GO) and functional information from literature, were identified in giraffe and okapi that were homologous to other mammals. This screen yielded 36 vision genes which were subjected to a series of analyses to determine selection pressure acting on these genes. Based on dN, dS and dN/dS parameters as estimated by the free-ratio model, no significant differences of the three evolutionary parameters

were observed between giraffe and okapi. In both species, overall dN, dS and dN/dS for each of the 36 vision genes were lower than 0.005, 0.05 and 0.1, respectively, suggesting that vision genes have evolved under strong purifying selection as expected.

The branch-site test of positive selection was performed on all 36 genes and identified lumican gene (*LUM*) as the only candidate exhibiting positive selection among the 36 vision genes in the okapi lineage (Fig. 1A). Aside from *PRPH2* and *CYP27B1* which had previously been shown to be candidates for adaptive evolution in giraffe (*Agaba et al., 2016*), no additional candidates of positive selection were identified in the giraffe lineages by the present analysis. Substitution analysis shows that the majority of sites (>80%) are conserved between okapi and closely related cetartiodactyl mammals (Fig. 1B). Positive selection in okapi's *LUM* is predicted to occur at a single codon site, GCG, at position 36 which encodes Alanine (A). This site exhibit strong posterior probability (0.94). The corresponding codon position in giraffe is AGA while in other species is AGG both of which encode Arginine (R). Besides the strong BEB posterior probability associated with codon position 36, there is also a peculiar observation that R36A substitution seems to have required at least two substitutions in the lineage leading to okapi.

To further evaluate the mechanisms of *LUM*'s codon 36 evolution and obtain context of the changes leading to 36A in okapi, ancestral *LUM* sequences at the interior nodes were reconstructed using a broader vertebrate phylogeny. Apparently, only five unique amino acid replacements at codon 36 were found in the whole vertebrate tree (Fig. 1B). Two of these changes occur deep within the phylogeny while three involve terminal nodes and their immediate ancestors. The common ancestor of all vertebrates appears to possess Proline (P, encoded by CCA) at this position. The most ancient nonsynonymous change leads to Glutamine (Q, CAA) in the common ancestor of mammals, followed by another nonsynonymous step leading to R (CGA) in the ancestor of ungulates. For the terminal taxa, convergent replacements R36A and Q36A are respectively observed in okapi and the Malayan colugo (*Galeopterus variegatus*) (Fig. 1C). Closer inspection of okapi and colugo protein sequences did not reveal other convergent changes (*Zhang & Kumar, 1997*). But using a branch-site test, a hypothesis of adaptive evolution on convergent site possibly due to similar selective pressure in both lineages was found to be significant ($P < 0.001$).

## Divergent selection pressure has shaped the evolution of giraffe and okapi important vision genes

The branch-site model assumption that adaptive evolution is determined by the presence of positive selection in foreground branches ignores the possibility that significant variation in $\omega$-ratios among all branches and sites may also be indicative of adaptive pressure (*Schott et al., 2014*). For the vision genes studied here, it is possible that specific branch(es) on the tree exhibit sites variation due to selection divergence. Upon examination of selection divergence using clade model C of *Bielawski & Yang (2004)* comparing giraffe or okapi and other ruminants in twenty genes critical to light transmission and light signaling processes, significant results were obtained for three genes while the rest were found to be not significant. Significant genes were *SAG* (found divergent between okapi and other

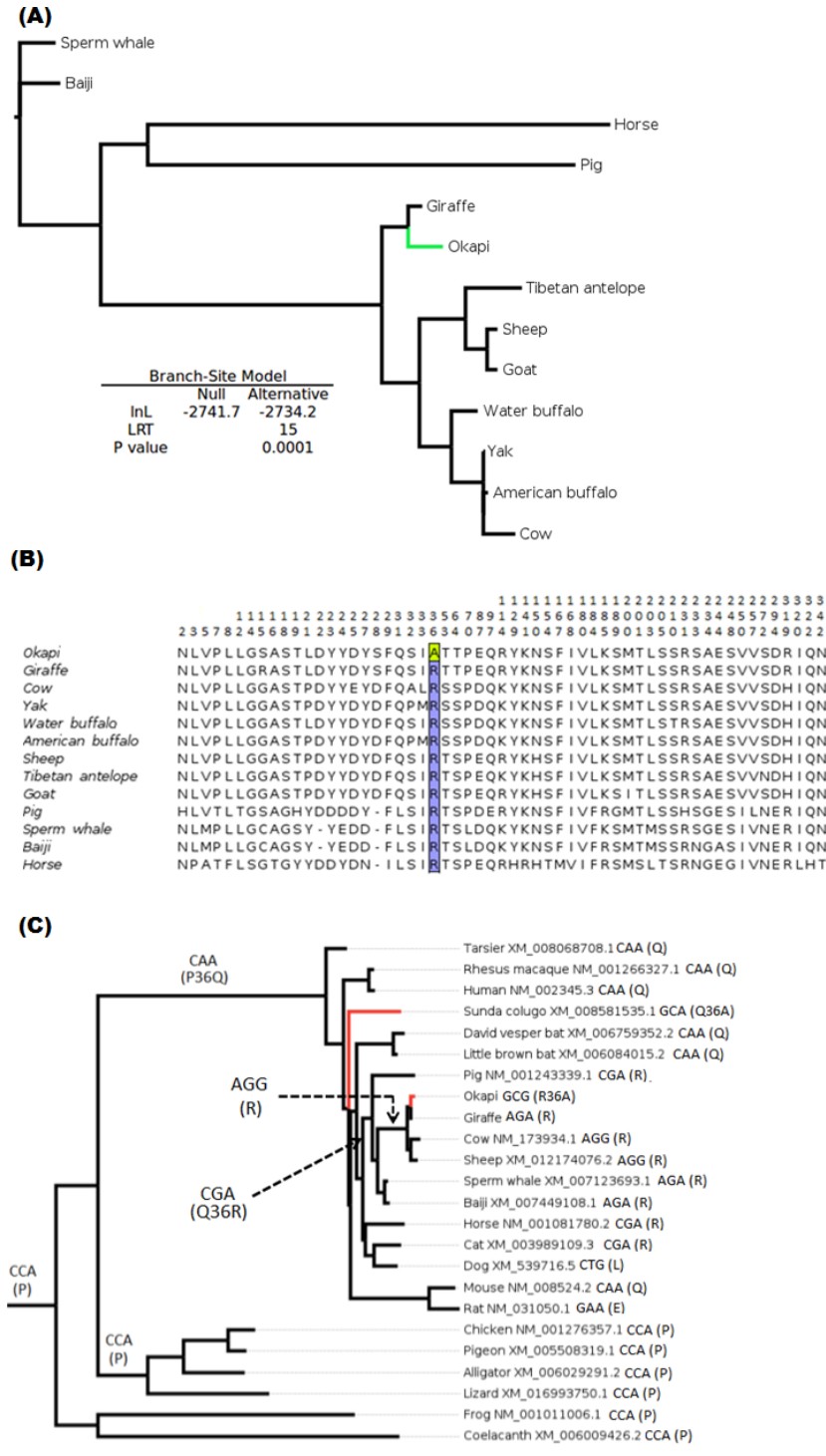

**Figure 1 Positive selection in *LUM* is predicted to have occurred in okapi (adapted to deep-forest) when compared to other ruminants inhabiting light illuminated environment.** (A) PhyML generated maximum likelihood *LUM* gene tree that was used in branch-site test for positive selection setting okapi as a foreground lineage. (B) *LUM* protein alignment showing positions at which okapi differ with species within ruminant, cetacean, equine and pig families. Conserved positions are 
**Figure 1 (…continued)**
omitted from the alignment. The codon position predicted to have undergone positive selection is color highlighted at site 36 which encodes a unique amino acid in okapi compared with other species in the alignment. (C) Vertebrate-wide evolution at codon 36 shows signature of convergent evolution between okapi and Malayan colugo (*Galeopterus variegatus*). The identity of codon at the predicted positive selection site and the respective amino acid (in bracket) are shown for each of the terminal species and for some ancestral lineages.

ruminants), *CRYAA* and *OPN1LW* (found divergent between giraffe and other ruminants) (Table 1).

In all three significant cases, vast majority of the sites (about 95%) were under strong purifying selection in both foreground and background lineages to keep their functions, while the proportion of divergent site classes was about 5%. The proportions of neutrally evolving sites were negligible. Notably, divergently evolving sites were under stronger purifying selection in *SAG* and *CRYAA* in the foreground lineages than in the background lineages (Figs. 2A and 2B). Furthermore, several residues in *SAG* and *CRYAA* at some divergently evolving sites are observed to be shared between giraffe and okapi (Figs. 2A and 2B).

For the divergent site class in *OPN1LW*, a remarkable case of rate acceleration was observed in the foreground lineage ($\omega = 339.6$) compared with the background lineages. Because it is theoretically possible for novel functions to be associated with selection divergence in orthologous genes, we next identified sites predicted to have high (>0.75) posterior probability score as determined by PAML's Bayesian computation. According to the five-sites rule, substitutions involving Serine (S), Alanine (A), Tyrosine (Y), Histidine (H), Phenylalanine (F) and Threonine (T) at sites 180, 197, 277, 285 and 308 of the L-opsin exert cumulative change in spectral shifts. In particular, the S180A, H197Y, Y277F, T285A and A308S substitutions modulate absorption spectrum by decreasing 7, 28, 7, 15 and 16 nm, respectively, from the maximum wavelength in an additive manner (*Yokoyama & Radlwimmer, 1999*). The reverse substitutions increase the maximum absorption spectrum by the same measures. Significant posterior probability scores in *OPN1LW* were found at amino acid site 180 and 233, which correspond with A180S and S233T. These substitutions are unique in giraffe when compared with other ruminants (Fig. 2C). Except for the A180S substitution, the other residues in the context of five-sites rule are identical between giraffe and other ruminants. The second giraffe specific substitution (S233T) also occurs at another spectrally important site within transmembrane domain 5 of the red pigment (Fig. 2D), where the A233S substitution has been observed to shift the wavelength by 1 nm (*Winderickx et al., 1992*).

To gain further insight into the functional significance of giraffe's *OPN1LW* selection divergence, we examined *OPN1LW* across broad range of mammals for possible functional convergence associated with the five critical sites. It can be observed that the entire *OPN1LW* gene tree is faithfully concordant with species phylogeny (Fig. 3). An inspection of codons corresponding to sites 180, 197, 277, 285 and 308 of *OPN1LW* shows similarities in the allelic combination of giraffe amino acids at the respective sites with pinnipeds, bats and some primates (Table 2). In these taxa, the overrepresented allele at the five sites is S,

Ishengoma et al. (2017), *PeerJ*, DOI 10.7717/peerj.3145
**Table 1** Significant selection divergence in three vision genes between giraffe or okapi (Clade 1) against the background of ruminant species (Clade 0).

| | Giraffe | | | | | | | | Okapi | | | | |
| | lnL | | | Site classes | | | | lnL | | Site classes | | | |
| Gene | M2a_rel | CmC | LRT | 0 | 1 | 2 | *P*-value | CmC | LRT | 0 | 1 | 2 | *P*-value |
|---|---|---|---|---|---|---|---|---|---|---|---|---|---|
| *CRYAA* | $-936.5$ | $-933.8$ | 5.3 | $P_0 = 0.9$ $\omega_0 = 0.0$ | $P_1 = 0.0$ $\omega_1 = 1$ | $P_2 = 0.1$ $\omega_{Clade\,0} = 1.4$ $\omega_{Clade\,1} = 0.0$ | 0.02 | $-935.1$ | 2.8 | $P_0 = 0.9$ $\omega_0 = 0.0$ | $P_1 = 0.0$ $\omega_1 = 1$ | $P_2 = 0.1$ $\omega_{Clade\,0} = 1.2$ $\omega_{Clade\,1} = 0.0$ | 0.09 |
| *SAG* | $-2177.5$ | $-2176.8$ | 1.6 | $P_0 = 0.4$ $\omega_0 = 0.05$ | $P_1 = 0.1$ $\omega_1 = 1$ | $P_2 = 0.5$ $\omega_{Clade\,0} = 0.5$ $\omega_{Clade\,1} = 0.2$ | 0.2 | $-2175.5$ | 4.1 | $P_0 = 0.95$ $\omega_0 = 0.08$ | $P_1 = 0.0$ $\omega_1 = 1$ | $P_2 = 0.05$ $\omega_{Clade\,0} = 0.0$ $\omega_{Clade\,1} = 2.3$ | 0.04 |
| *OPN1LW* | $-1780.6$ | $-1778.2$ | 4.7 | $P_0 = 0.96$ $\omega_0 = 0.0$ | $P_1 = 0.03$ $\omega_1 = 1$ | $P_2 = 0.01$ $\omega_{Clade\,0} = 0.0$ $\omega_{Clade\,1} = 339.6$ | 0.03 | $-1780.2$ | 0.7 | $P_0 = 0.95$ $\omega_0 = 0.0$ | $P_1 = 0.0$ $\omega_1 = 1$ | $P_2 = 0.05$ $\omega_{Clade\,0} = 0.9$ $\omega_{Clade\,1} = 0.0$ | 0.4 |

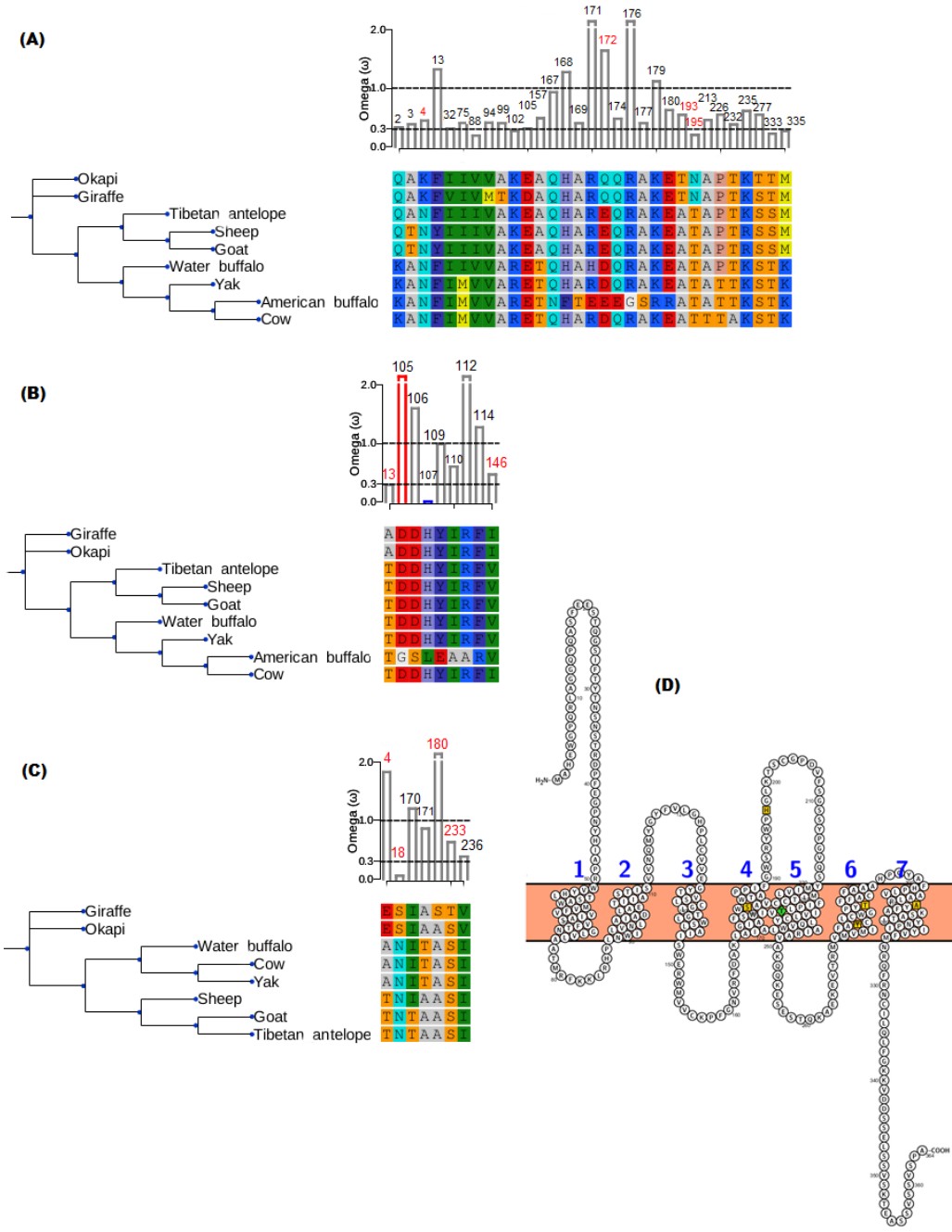

**Figure 2 Gene trees and substitution analysis indicating sites (highlighted in red) potentially contributing to selection divergence in *SAG, CRYAA* and *OPN1LW*.** The ω ratio for each variant site in the three genes was computed using the site-wise likelihood ratio analysis (*Massingham & Goldman, 2005*). (A) *SAG* has five sites which show signature of differential selection between okapi and other ruminants. (B) *CRYAA* has several sites which exhibit signature of selection but the associated sequence changes in giraffe are shared with okapi. (C) Substitution analysis shows seven variant sites (4, 8, 170, 171, 180, 233 and 236) which differ between giraffe and any ruminant species shown in the phylogeny. (continued on next page...)
**Figure 2 (…continued)**
Variant sites 180 and 233 have Bayes posterior probability of 0.93 and 0.89 respectively. All giraffe's sequences were found to be identical between NZOO and MA1 (*Agaba et al., 2016*) verifying that the identified substitutions are not artifacts. (D) The ribbon diagram of giraffe L-opsin highlighting important sequence changes relative to its secondary structure. Its spectrally important amino acids based on the five-sites rule are highlighted in yellow. A threonine at amino site 233 with respect to its unique S233T among ruminants is highlighted in green. The ribbon plot for the *OPN1LW* was generated by Protter (*Omasits et al., 2013*).

**Table 2 Identity of amino acids at the functionally important sites in the L-opsin of giraffe and other mammals based on the five-sites rule.** For each species, the expected $\lambda_{max}$ based on the five-site rule is shown. Where the actual $\lambda_{max}$ of the pigment has been determined, the value is indicated.

| Species | Site/amino acid | | | | | Predicted $\lambda_{max}$ (nm) | Experimental $\lambda_{max}$ (nm) | References |
|---|---|---|---|---|---|---|---|---|
| | 180 | 197 | 277 | 285 | 308 | | | |
| Human (*Homo sapiens*) | S | H | Y | T | A | 560 | 557 | *Merbs & Nathans (1992)* |
| Human (*H. sapiens*) | A | H | Y | T | A | 553 | 552 | *Merbs & Nathans (1992)* |
| Rhesus macaque (*Macaca mulatta*) | S | H | Y | T | A | 560 | 561.5 | *Bowmaker et al. (1991)* |
| Crab-eating macaque (*Macaca fascicularis*) | S | H | Y | T | A | 560 | 561 | *Baylor, Nunn & Schnapft (1987)* |
| Baboon (*Papio anubis*) | S | H | Y | T | A | 560 | 560 | *Bowmaker et al. (1991)* |
| Green monkey (*Chlorocebus sabaeus*) | S | H | Y | T | A | 560 | – | |
| Seal (*Phoca vitulina*) | S | H | F | T | A | 552 | 510 (?) | *Crognale et al. (1998); Levenson et al. (2006)* |
| Walrus (*Odobenus rosmarus*) | S | H | Y | T | A | 560 | – | |
| Brandt's vesper bat (*Myotis brandtii*) | S | H | Y | T | A | 560 | – | |
| David's vesper bat (*Myotis davidii*) | S | H | Y | T | A | 560 | – | |
| Big brown bat (*Epistesicus fuscus*) | S | H | Y | T | A | 560 | – | |
| Little brown bat (*Myotis lucifugus*) | S | H | Y | T | A | 560 | – | |
| Cow (Bos taurus) | A | H | Y | T | A | 553 | 555 | *Jacobs et al. (1998)* |
| Sheep (Ovis aries) | A | H | Y | T | A | 553 | 552 | *Jacobs et al. (1998)* |
| Goat (*Capra hircus*) | A | H | Y | T | A | 553 | 553 | *Jacobs et al. (1998)* |
| Giraffe (*Giraffa camelopardalis*) | S | H | Y | T | A | 560 | – | |
| Okapi (*Okapia johnstonii*) | A | H | Y | T | A | 553 | – | |

H, Y, T and A (henceforth denoted here as SHYTA) for sites 180, 197, 277, 285 and 308, respectively. Giraffe's SHYTA allele, although unique to ruminants, is observed in common with some humans, some old-world primates, walrus and vesper bats.

The pattern of allelic combination shows sites 197, 285 and 308 to be invariant in these taxa while site 180 shows some degree of S/A variation between- and within- species. In particular, humans are known to be polymorphic for the S/A allele at site 180 of *OPN1LW* (*Winderickx et al., 1992*). The SHYTA and AHYTA forms of human *OPN1LW* have their respective $\lambda_{max}$ experimentally determined by *Merbs & Nathans (1992)* at 557 nm and 552 nm. If the identity of residues at the five sites is a reliable predictor of spectral tuning (and several experiments suggest it is (Table 2)), then a 5 nm variation in the $\lambda_{max}$ of *LWS* pigments between giraffe and other ruminants is expected.

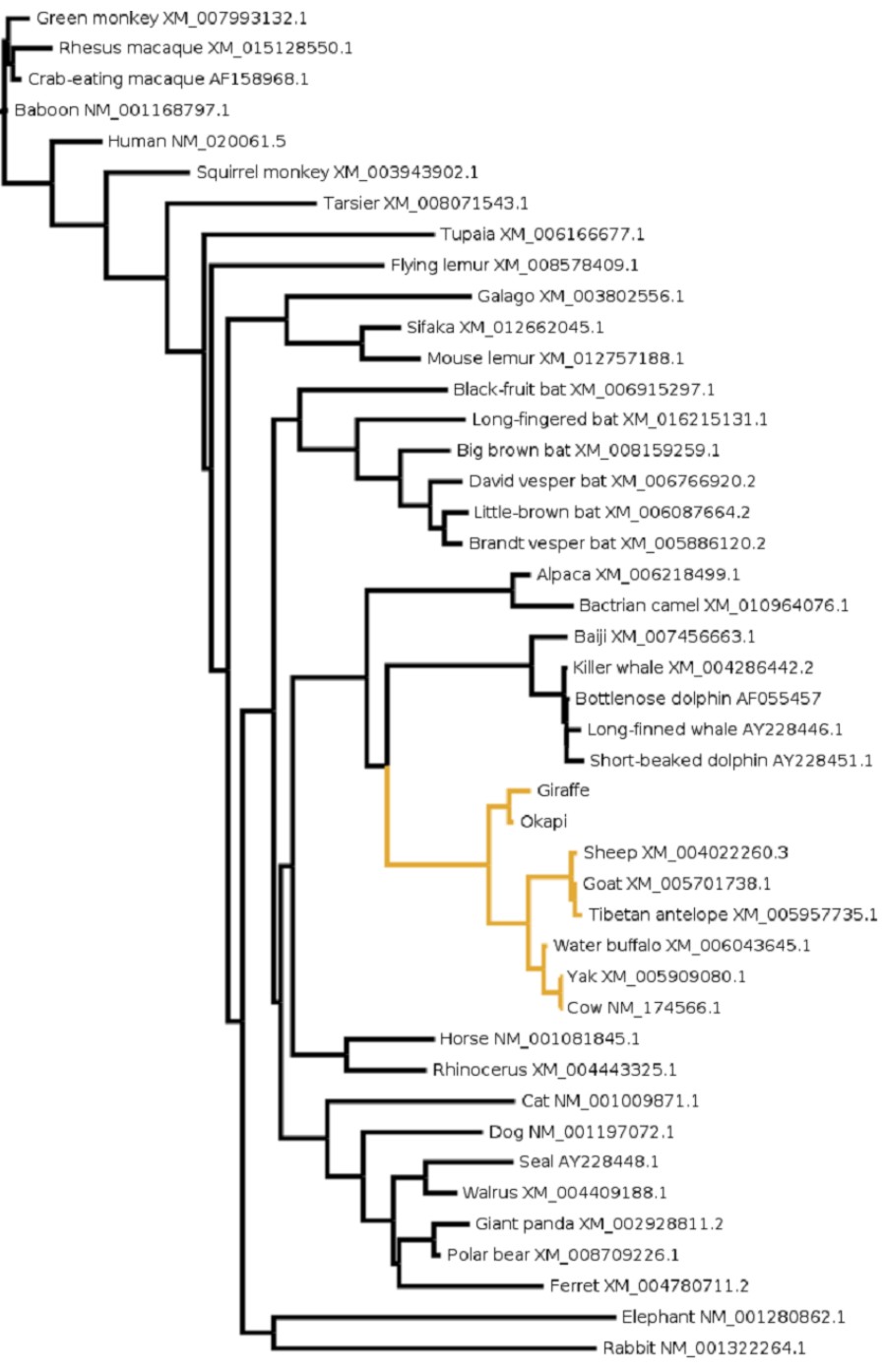

**Figure 3  The evolutionary relationship in mammals based on *OPN1LW* as revealed by coding gene sequences.** For species whose sequences were obtained from public database Refseq or Genbank accession numbers for the respective sequences are shown.

## DISCUSSION

The development of distinct attributes between species for a given trait is typically complex involving a large number of genes. The visual system is well known to be comprised of the actions of many genes, some of which exhibit tissue- and/or cell-type restricted functions (*Siegert et al., 2012*). In this study, we performed an evolutionary analysis on several genes participating in wide variety of visual process. The ultimate aim is to determine molecular genetic basis and evolutionary factors underlying giraffe excellent vision and its apparent disparity with that of okapi.

We discovered positive selection and selection divergence in genes with predominant roles in corneal, lens and retinal functions in both organisms. This suggests that the focal point of selection on the vision phenotype may not be limited to a single functional unit. Rather, the interplay of different elements in the visual pathway appears to be mirrored by the operation of natural selection on functionally diverse vision genes, possibly to adjust species' vision to the myriad of complexity of their visual environment. This potentially concerted adaptive strategy is likely partly due to the intricate functional matching between ocular and retinal aspects, a phenomenon perhaps well demonstrated in teleost fish and birds' visual systems. For example, in cichlids and birds, the degree by which the lens of particular species filters ultra-violet (UV) light is reflected by the levels of expression or possession of their UV pigments (*Hofmann et al., 2010*; *Lind et al., 2014*). Recent findings also suggest widespread differential capacities in ocular transmittance of UV light among mammals (*Douglas & Jeffery, 2014*). The fundamental question is whether a pattern of selection signatures detected in okapi and giraffe vision genes is specifically an adaptive response imposed by their visual environment, or whether it is a pattern specific for other purposes. Our study can only provide insight on the likely role of environment influences on the evolution of vision genes in giraffe and okapi.

Among okapi vision genes, *LUM* is shown to have undergone positive selection. The irradiance spectrum at the floor of the low-light forest environment in which the okapis are confined is expected to be dominated by shorter wavelengths in the blue–green, blue and even ultraviolet ranges (*Honkavaara et al., 2002*; *Warrant & Johnsen, 2013*). Recently, *Douglas & Jeffery (2014)* shows okapis to possess a higher degree of UV transmission through their ocular media than closely related artiodactyls living in the open environment. Although specific sites required for *LUM* interaction with collagens are not yet identified (but see *Kalamajski & Oldberg, 2009*), the probable site under positive selection shows signal of convergent adaptation with the nocturnal Malayan colugo (Fig. 1B). Okapi and colugo are evolutionarily diverse organisms but one thing that they have in common is their low light forest habitat and nocturnal lifestyles. In general, nocturnal mammals tend to have eyes adapted to transmit and process UV (*Zhao et al., 2009*; *Douglas & Jeffery, 2014*). It is therefore likely that the convergence of positive selection in *LUM* in okapi and Malayan colugo may be associated with ocular adaptations related to spectral transmittance and survival in reduced light environment.

Another gene found to exhibit signature of selection in okapi is *SAG*. Recently, *SAG* was found to show strong evidence of convergent evolution in species adapted to dim-light

vision (*Shen et al., 2012*). Arrestin binds to photoactivated and phosphorylated rhodopsin, a process that desensitizes rhodopsin and regulates the signaling process. The mutation in the gene causes congenital stationery night blindness and other retinal diseases (*Kuhn, Hall & Wilden, 1984*; *Fuchs et al., 1995*; *Nakazawa, Wada & Tamai, 1998*). It might be that evolutionary changes in *LUM* (associated with corneal transparency) and *SAG* (important in rod mediated vision) could confer okapi with concerted adaptive mechanisms associated with requirement for low light vision and exploitation of the deep forest niche.

Besides *PRPH2* and *CYP27B1* (*Agaba et al., 2016*), this study further identifies selection divergence in *CRYAA* and *OPN1LW* between giraffe and other ruminants. Observed amino acid changes associated with selection divergence in giraffe's *CRYAA* are shared with okapi and in some cases with other ruminant species (Fig. 2B). This may suggest common phylogenetic signal or shared functional variation. In the latter case, we would expect the rate of molecular evolution on *CRYAA* to be determined by the extent of selective constraint associated with maintenance of vision among species. Mammals such as the subterranean mole rat exhibit an accelerated rate of *CRYAA* evolution at nonsynonymous positions when compared with the visually oriented rodents, consistent with the mole rat little need for vision (*Hendriks et al., 1987*). Since variation in visual qualities are observed among artiodactyls (*Veilleux & Kirk, 2014*), it is reasonable to speculate that divergence in selective pressure on *CRYAA* may reflect the degree of reliance on αA-crystallin functions in the lens associated with relative visual requirements of giraffe and other ruminants.

But a comparison of giraffe's *OPN1LW* with those of other ruminants identifies two unique changes that could provide giraffe with unique color-based tuning (Fig. 2D). The first change is the S233T which occurs in TM5 of the receptor (Fig. 2D). The location of the change is potentially important for the "chloride effect", a phenomenon observed to be widespread in vertebrate L-cone pigments (*Wang, Asenjo & Oprian, 1993*; *Hirano et al., 2001*). In vertebrates, the binding of chloride ion at H197 in both L- and M-cone pigments contributes to the red-shift of the chromophore's absorption in the visual pigments. Whatever shift caused by S233T in the L-cone spectral tuning, it is most likely to be subtle as both Serine and Threonine are hydroxyl-bearing amino acids (*Merbs & Nathans, 1993*). Establishing actual impact of S233T in the spectral tuning of the L-cone pigment requires functional studies.

The second change is the A180S substitution at one of the five functionally significant sites of the red opsin. This confers giraffe with an SHYTA allele compared with an AHYTA allele observed in okapi and other ruminants in the study. Based on the five-sites rule, this is expected to provide giraffe with at least 5 nm spectral-shift toward red when compared with other ruminants (*Yokoyama & Radlwimmer, 1998*; *Yokoyama & Radlwimmer, 1999*). It should be noted, nevertheless, that the two giraffe individuals sequenced in the project are insufficient to conclude that A180S substitution is fixed in the giraffe populations. Hence, the site may as well be polymorphic in giraffes as in humans.

If sequence changes observed in giraffe's *OPN1LW* confer giraffe with adaptive benefits over other ruminants, these advantages can only be speculated upon based on some common environmental challenges with other species with which it shares the SHYTA genotype. This is based on the key assumption that distantly related species facing common

Peer J

or related problems are likely to respond by developing convergent solutions (*Stern, 2013*). It is, therefore, notable that giraffe share the SHYTA genotype with distantly related mammals such as pinnipeds, bats and some primates (Table 2). This may reflect some functional convergence associated with *LWS* color discrimination among these species.

Pinnipedia (seals and walruses) are marine mammals whose existence depends upon switching their lives between land and shallow coastal waters. In coastal waters, where seals and walruses spend majority of their time, are dominated by longer wavelength colors such as green and red (*Wozniak & Dera, 2007*). Because of their amphibious lifestyle, seals and walruses are presented with the need to adapt to the dominant spectral irradiance when in water and also maintain some color vision when on land (*Griebel & Peichl, 2003*). Apparently, pinnipeds, in their course of evolution, have become cone monochromats after losing functional S-cone pigment and retaining only L-cone pigment (*Crognale et al., 1998*). Presence of five-sites SHYTA genotype in pinniped L-cone pigment could provide optimal ability to contrast rod and cone signals necessary for residual color vision they may possess (*Crognale et al., 1998*; *Griebel & Peichl, 2003*).

The importance of red color vision in bats is not clear but some bat species, including fruit-eating bats, possess intact, functionally constrained *OPN1LW* gene that, on its own, may provide bats with spectral mechanism for increased sensitivity during navigation and foraging (*Wang et al., 2004*). Alternatively, since bats are nocturnal mammals, their red color sensitivity could depend on the Purkinje shift. The Purkinje effect is the apparent reduction in brightness of red object under dim light conditions as a result of comparison of signal generated by the $\lambda_{max}$ of rhodopsin and that of the *MWS/LWS* opsins (*Trezona, 1970*). Under conditions where bats might benefit from signal inputs of both cone and rod systems, for example under twilight at dusk or dawn (*Pavey et al., 2001*; *Melin et al., 2014*), the SHYTA genotype may provide a spectral mechanism to compensate for the Purkinje effect.

Anthropoid primates (consisting of New World and Old World monkeys) are usually regarded as visual specialists due to their keen visual acuity and color perception (*Kirk & Kay, 2004*; *Kawamura, 2016*). Indeed, the best example in higher mammals linking sequence variation in L/M opsin to visual behavior in an ecological context is observed in primates. In ateline New World primates which exhibit allelic variation at three of the five spectrally important sites in the L-opsin (where SYT, SFT, AFT and AFA at site 180, 277, and 285, respectively, are common variants), possession of SYT is likely advantageous among primates in identifying ripe fruits in the background of green leaves (*Matsumoto et al., 2014*).

As a general rule, matching of cone pigments spectral characteristics to spectral reflectance of the visual environment (*Lythgoe, 1984*), should provide clues to ecological factors driving visual genetic adaptations in species. For example, all woolly lemurs (*Avahi spp*) endemic to Madagascar forests uniformly possess cone pigment allele with $\lambda_{max}$ precisely tuned to the spectral reflectance of the preferred diet of young leaves (*Veilleux et al., 2014*). Likewise, diversification in the cone pigment spectral sensitivity in fish species influences detection of prey and communication with conspecifics (*Sabbah et al., 2010*). But the suite of ecological factors more likely to play an important role in vision
adaptations is shared among ruminants and other artiodactyls of Savannah. This makes the sequence changes in giraffe *LWS* opsin rather interesting and poses challenge in applying conventional ecological arguments to the finding. Although the expected magnitude of $\lambda_{max}$ variation caused by the A180S is quite small ($\sim$5 nm), the presence of an additional change at site 233 located in TM5 of L-opsin suggests that the gene is under differential selective pressure in giraffe. Since polymorphisms in human *OPN1LW* are likely adaptive (*Verrelli & Tishkoff, 2004*; *Verrelli et al., 2008*), it is reasonable to also postulate on selection drivers acting on *OPN1LW* in the giraffe.

*Mitchell et al. (2013)* and others have speculated that giraffe height and extraordinary visual capacity may have co-evolved. Gradual acquisition of long necks may have provided giraffes with selective ability to see predators from afar, besides the feeding advantage for nutritious and top placed foliage (*Mitchell & Skinner, 2003*; *Williams, 2016*). Indeed, because of their excellent aerial vision, adult giraffes are rarely killed by lions (*Periquet et al., 2012*; *Strauss & Packer, 2013*). But giraffe height advantage to see lions from afar likely presents challenges in identifying camouflaged lions in the background of tall dry grass of the semi-arid Savannah. This probably explains the overrepresentation of giraffes among lion kills during the arid season (*Owen-Smith, 2008*; *Davidson et al., 2013*). Therefore, sequence changes in giraffe's L-cone pigment could be an adaptive response to predation pressure that provides giraffes with differential spectral mechanisms for enhanced ability to discriminate between dry savannah vegetation and lions.

The alternative explanation is intra-specific communication among giraffe populations. Among major giraffe populations, the characteristic pelage color is reddish-brown spots separated by variable network of fine white lines (*Brown et al., 2007*; *Fennessy et al., 2016*). In fish, avian and other mammal species, patches of color on the face or tails are clearly for communication purpose with conspecifics (*Caro, 2005*; *Price et al., 2008*; *Stoddard & Prum, 2011*). Since various giraffe subspecies exhibit polymorphisms in pelage pattern and often have overlapping home ranges, it is thought that one of the ways giraffes maintain reproductive isolation in the wild is through pelage-based mate discrimination (*Brown et al., 2007*). Thus, one might think that spectral tuning of the giraffe visual pigments should also match with characteristic color of conspecifics and allow for easy recognition among giraffes. Accordingly, giraffe's L-cone pigment might possess a slightly red-shifted $\lambda_{max}$ to enable sharp detection of conspecifics for several purposes, including preferred mate choices.

At the present time there shouldn't be *a priori* reasoning that one or the other argument is a correct explanation of the actual significance of adaptive evolution in giraffe or okapi vision genes. For example, there is currently no evidence on whether sequence changes in giraffe's L-cone pigment contribute sufficiently or not in conferring variation in the ability to detect predators or discriminate conspecifics. Direct evidence to support these speculations necessarily requires studies linking sequence evolution to functional changes. Towards this goal, first, the fixation status of the sequence changes needs to be clarified among giraffe and okapi populations. Secondly, a combination of divergence-based and population genetics approaches is potentially suited to ascertain the adaptive consequences of selection divergence (*Verrelli et al., 2008*). Finally, definitive link between sequence

changes and adaptation require ecological and comparative studies on giraffe and okapi visual behaviors.

## CONCLUSIONS

The subset of genes known to play functional role in vision has been analyzed in order to identify if remarkable differences in vision between giraffes and okapi are associated with adaptive evolution. The discovery that visual genes are highly conserved in their evolution signifies strong purifying selection in giraffe and okapi visual genes. Putative evidence of positive selection and selection divergence is observed on few candidate vision genes in both giraffe and okapi. Signature of selection in genes functionally associated with important optical elements of the eye, such as the cornea, the lens and the retina, could be indicative of concerted, organ-level impact of natural selection in adjusting species' vision to their respective environment. This demonstrates the importance of system-level understanding of molecular evolution associated with complex traits (*Invergo et al., 2013*). We believe that comparative evolutionary vision studies such as this could contribute to the understanding of the molecular genetic system underlying vision in mammals in general.

### Funding

This work received support from the Tanzania Commission of Science and Technology, Nelson Mandela African Institute of Science and Technology, Penn State University, Biosciences Eastern and Central Africa–International Livestock Research Institute, Nashville Zoo and White Oak Holding and SEZARC. The funders had no role in study design, data collection and analysis, decision to publish, or preparation of the manuscript.

### Grant Disclosures

The following grant information was disclosed by the authors:
Tanzania Commission of Science and Technology.
Nelson Mandela African Institute of Science and Technology.
Penn State University, Biosciences Eastern and Central Africa–International Livestock Research Institute.
Nashville Zoo and White Oak Holding.
SEZARC.

### Competing Interests

The authors declare there are no competing interests.

### Author Contributions

- Edson Ishengoma conceived and designed the experiments, performed the experiments, analyzed the data, wrote the paper, prepared figures and/or tables, reviewed drafts of the paper.

- Morris Agaba and Douglas R. Cavener conceived and designed the experiments, performed the experiments, analyzed the data, contributed reagents/materials/analysis tools, reviewed drafts of the paper.

## Data Availability

Genome data in which coding sequences for giraffe and okapi vision genes were extracted is publicly accessible in the Short Read Archive under project number SRP071593 (BioProject PRJNA313910). Accession numbers for the sequences from other species used in this study are presented in Results as part of the manuscript in Figs. 1C and 3, and as supplemental files in Files S3 and S4.

## Supplemental Information

Supplemental information for this article can be found online at http://dx.doi.org/10.7717/peerj.3145#supplemental-information.

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
