# Peer review of "Evolutionary analysis of vision genes identifies potential drivers of visual differences between giraffe and okapi"

_PeerJ, doi:10.7717/peerj.3145_

## Round 0.1 · original submission · Major Revisions

This study has now received two thorough reviews from experts in the field of comparative genomics, including the evolution of vision. Both Referees thought this was an interesting and well designed study. Nonetheless, I agree with the Referees that revisions in association with two main caveats need to be addressed. First, I agree with the main concern from both Referees that the paper is difficult to follow because of the quality of the writing. I trust the authors will endeavour to make significant improvements to the grammar with major revisions. Second, both Referees indicated that there are often assumptions left unstated (e.g. evolution of LUM) and in some cases this may contribute to conclusions not supported by the data. The reviews make suggestions on where this could be improved. In the revised version, please address each and every comment made in these helpful reviews.

Reviewer 1 ·

Basic reporting

No big concerns. English is perhaps not the authors first-language, and some clean-up of grammar is warranted. Sentence structure can be improved/ split; for example:

• LINE 74-76: Giraffes have excellent aerial vision reinforced by their long necks, which is uniquely the highest among ruminants and predominantly rely on vision communication relative to other senses.
o Giraffes predominately rely on visual communication due to their excellent aerial vision. This is partly mediated by their uniquely long necks in comparison to other rudiments.

• Figure format: LUM positive selection
o LINE 703: Potentially use color coding alignment software (JalView or similar), to better illustrate subtle residue changes.

Line 265-276 could be better presented as a Table, allowing inclusion of accession and/or Ensembl numbers, or maybe some stats on %id amongst the key species of interest

Line 293 – provide a reference for your past work

Experimental design

Line 295 – it is unclear how LUM became an object of focus for the first result. Perhaps clarify by beginning with a broader summary; e.g. was this the only gene that showed positive selection; if not, why focus on it and if yes this adds some rigor and intrigue to the finding. Did this LUM result come out of the genome-wide scan or the ad hoc list of vision genes? I suspect it is the former, as this is not a well-known vision gene (not in the list of the top 200 genes I’d name ‘off the top of my head’; indeed I’ve never heard of it and claim no omniscience).

What is really unclear to me, looking at the alignment of Lum orthologs in Figure 1b, is why codon 36 is empirically more interesting than codons 15 or 64. These both also seem to show dramatic selection. I am not an expert in the method: perhaps the variance at codon 15 makes this uninteresting; T64S is a trivial change; in any case giving the reader a bit more explanation may be in order if you want a broader audience to appreciate your Results.

Having identified Lum^R36A as being rare, an easy go-to check of its veracity would be to ask how this residue changes (or not) in a broader suite of vertebrate lineage.


Line 311 – describe something about the other 17 genes. I guess they were non-significant; is this only because a limited species set was used? Or are these genes clearly not undergoing adaptive selection in this clade?

Claim: OPNLW1 shows significant divergent selection in giraffe compared to other ruminants, suggesting a novel visual adaptation (shift to higher red sensitivity).

LINE 407: A180S in giraffes OPNLW1
• How different is 5 nm, is this actually biologically relevant? Need to mention that we have NO way of knowing if this is meaningful at a processing level.

• Discussion alludes to differentiating between lions and open savannah in long distance areas but gives no support… Stating some functional differences between perception of wavelengths would be helpful to support this conclusion.

LINE 732: FAILED TO ELABORATE ON FINDING (Humans are polymorphic at 180 for A/S).
• Could mention that giraffe may also naturally contain this variation. It is a newly sequenced genome from only two individuals, and although both individuals had A180S, we cannot take this as “always true” LINE 710.

LINE 744: FAILED TO ELABORATE ON ASSUMPTION: Giraffe peptide sequence closer to seal, walrus, bats and new-world monkeys when aligned with five site residues.
• Is it biologically relevant to group organisms based on specific coding residues- yes they have been deemed important in spectral sensitivity, but aren’t we losing a lot of information if we group them like this?
• Assumes if this does tune spectral sensitivity and this is unique to giraffes, they share this with other organisms who have completely unrelated niches, diets and predators (ie. Seal, walrus and bats).
LINE 421: FAILED TO ELABORATE ON ASSUMPTION: Bats have similar genotype at these five residues, and it may help them see ripe fruits (red) when they are hunting.
• Alternative reasoning: Bats are mostly active at night; they may need boost into red spectrum because of Purkinje effect (shift into blue spectra at low levels of illumination). Yes, this is a speculation that bats are similar to human, but must be able to defend alternative hypotheses.
• Seals/walrus are marine creatures where red wavelengths are lost first at depth- make sense why they could be more sensitive.
• Giraffes are mostly active during the day… Why don’t opaki exploit this as well, as they could benefit from increased red sensitivity in low-light conditions.

Validity of the findings

It seems like there are often assumptions left unstated. e.g. it is a major assumption that the LUM gene is receiving selection pressure for its role in the visual system, when its been product is expressed in other tissues. LUM is in ligaments, and I think it is fair to say that giraffes are infamous for having unusual morphology that might account for this selection.
The approach doesn't really allow one to assess robustness, or statistical validity. It is quite descriptive.

Reviewer 2 ·

Basic reporting

My main concern is the writing of the paper. Throughout the introduction (and beyond) there are many run-on sentences, missing words, and unclear phrasing that make the paper hard to read. In some sections, almost every sentence is difficult to decipher. The paper needs major revision of the writing, and should be carefully proofread by an expert in English writing and grammar.

A non-complete list of some examples of wording/phrasing or problems with language.
Lines 79-80: missing word in this sentence?
Lines 85-86: Missing words?
Lines 86-87: Missing words?
Lines 88: converts
Line 128-131: Example of very poorly written sentence.
Lines 181-183: Missing words? I am not sure what this is saying.
Lines 250-252: Missing words?
Lines 265-272: Better as a table.

In addition to the difficulty with writing conventions, the structure of the introduction should be revised. As it is, the most of the introduction is on the structure (anatomy) and function (processing) of the visual system. These components are not directly relevant to the study, and are not revisited in the discussion or interpretation of the results. By focusing on these aspects of visual systems, the genetics underlying vision is lost. It would be stronger to start the introduction with the genetic basis of visual systems and then introduce the system, previous findings and therefore set up the current question/objective.

The authors should make links to the common conventions of the gene classes. SWS/MWS/LWS are often used to describe the opsin genes and including reference to those classes in addition to the OPN1LW terminology will be helpful to readers that are familiar with visual ecology in non-mammalian systems.

The in-text citations are not consistent in their use of a ‘,’ after the authors and before the year.

Experimental design

The objective is clear and the study seems well designed. Some of the details may be made more clear from editing and proofreading of the paper, as the writing issues did continue throughout the methods and results.

I am not sure how the analysis of the tuning sites themselves in a phylogenetic framework (figure 3b) adds to the objective of the paper. It is interesting and important to identify those species that share the same visual system, but a phylogenetic analysis is not necessary to do so.

Validity of the findings

Related to a point in the experimental design, the discussion of the shared SHYTA genotype should be expanded. It currently does not fully address the evolutionary pressures in all organisms that share the type (pinnipeds are not discussed). Further, the likely independent evolution of this visual genotype is particularly interesting and exciting. Focusing on that as an umbrella to discuss the known (or unknown) functions in the other species could strengthen the discussion as it relates to this system.

There is no discussion of the CRYAA gene in terms of functional adaptive significance. This should be added so that all the genes of importance and discussed in detail.

In Discussion, a broader context of adaptation to visual systems should be incorporated. For example, fish systems are some of the best studied and most well-known for work on environment based visual adaptations. These can provide insight to the interpretation of the results.

Lines 349-54: Should be clear about if this site is/is not in a transmembrane region of the opsin protein. This will further allow for interpretation of if it may be spectrally important.

Additional comments

This is an interesting, well designed and well executed study. However, the manuscript will be stronger and easier to read after revision.

---

## Round 0.2 · Minor Revisions

This revised MS has dealt with many of the concerns that were raised, namely edits for clarity with respect to the assumptions in the paper. Nonetheless, I tend to agree with the Reviewer that some balance in the discussion of the putative adaptive significance be addressed. In addition, this Reviewer made some excellent suggestions to balance the speculations in the discussion based on the data. These concerns should be easy to address with minor edits.

Reviewer 2 ·

Basic reporting

Please proofread the abstract. The paper is much improved in language, however the abstract was not edited and for example, the first sentence of the abstract is missing ‘to’ and says ‘respond visual signal’

Line 176: insert ‘candidate’ to ‘To obtain vision genes’
Lines 185-186: change to ‘searches of the literature for proof’
Line 217: change to ‘were used as guides for’
Line 387: change to ‘on vision phenotypes’ or ‘on the vision phenotype’
Line 464: get rid of ‘in’

Experimental design

The experimental design appears sound.

Validity of the findings

A previous comment stands: There is no discussion of the CRYAA gene in terms of functional adaptive significance. This should be added so that all of the genes of importance are discussed in detail.
It is now mentioned in lines 431-433, but there are pages of interpretation of LUM and OPN1LW so it seems odd that CRYAA is not discussed in depth.

Lines 511-523: This seems like a big stretch in terms of interpretation. Is there any evidence that a small shift in wavelength sensitivity would alter ability to distinguish camouflaged lions?

Lines 524-535: similarly, is there any evidence that giraffes vary in color? What is the interpretation of its function in color-related recognition based on?

Additional comments

A meaningful discussion of the significance of CRYAA is still missing. Also, there are parts of the discussion which are overly speculative and should be revised (Lines 511-535).

---

## Round 0.3 · accepted · Accept

In this version the conclusions are more balanced with the data and it reads really well. The study will be of interest to many in evolutionary biology and those studying vision. Thank you for sending your interesting paper to PeerJ.